# Experimental Study on Shear Behavior of Precast High-Strength Concrete Segmental Beams with External Tendons and Dry Joints

Zebin Hu [1], Zhenming Xu [2], Shufeng Zhang [2], Haibo Jiang [1,*], Yuanhang Chen [1] and Jie Xiao [1]

[1]  School of Civil and Transportation Engineering, Guangdong University of Technology, Guangzhou 510006, China; 2111909076@mail2.gdut.edu.cn (Z.H.); chenyuanhang@gzmtr.com (Y.C.); xiaojie2017@gdut.edu.cn (J.X.)
[2]  Guangzhou Communication Design Institute Co., Ltd., Guangzhou 511400, China; xuzhm002@vip.163.com (Z.X.); zsf13570481603@163.com (S.Z.)
*  Correspondence: hbjiang@gdut.edu.cn

**Abstract:** Precast high-strength concrete segmental beams with external tendons and dry joints (ED-PHCSBs) have become a potential alternative for achieving accelerated bridge construction due to their lighter self-weight and easier installation. In order to investigate the shear behavior of ED-PHCSBs, eight precast concrete segmental specimens were fabricated and tested to failure. For comparison purposes, one externally prestressed high-strength concrete monolithic beam was also investigated. The primary parameters, including concrete strength, shear span-depth ratio, stirrup ratio, joint number, and joint location, were adopted. Test results indicated that increasing the concrete strength or stirrup ratio can effectively improve the shear capacity of the ED-PHCSBs. The shear span-depth ratio was inversely proportional to shear strength for all specimens. The results also revealed that the joint number had a marginal effect on the defections and stresses of the external tendons of ED-PHCSBs. AASHTO 2017 and Chinese code 2018 can conservatively estimate the shear strength of ED-PHCSBs. Considering the actual failure modes of the precast beams, a calculation method based on a modified strut-and-tie model was proposed. The average and standard deviation of the ratios of the test results to the predicted value of the proposed method were 0.98 and 0.08, respectively. It indicated that the proposed formula was more accurate.

**Keywords:** precast high-strength concrete segmental beams; shear behavior; strut-and-tie model; external tendons; dry joints

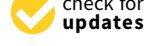



## 1. Introduction

For more than 40 years, precast concrete segmental bridges (PCSBs) have been constructed widely, owing to the demand for their economical design and increased speed of erection. These bridges have also been recognized as a solution to many bridge problems due to improved quality control, mitigation of traffic disturbances, and superior durability.

The first application of PCSBs was Long Key Bridge [1]. Bang Na Expressway in Bangkok also exhibited many advantages of PCSBs [2]. Recently, several famous projects of PCSBs have been built in China, such as Sutong Yangtze River Bridge, Nanjing No.4 Yangtze River Bridge, Zhuhai-Macau Lotus Bridge [3], and Humen Second Bridge [4]. Although the joints between precast concrete segments are usually epoxy joints to prevent the corrosion of internal prestressing strands, dry joints are becoming popular due to their rapid construction, lower costs, and elimination of additional curing [5–8]. Dry joints located between concrete segments represent locations of discontinuity in PCSBs, which influence the overall structural behavior of PCSBs, including the ultimate flexural capacity or shear strength. When dry joints were adopted in bridge design, internal tendons were prohibited by the (American Association of State Highway and Transportation Officials) AASHTO 2003 [9], and external tendons were the only alternative.

However, the combined use of external tendons and dry joints in PCSBs significantly affects the shear behavior of PCSBs. Dry joints are the weakness of PCSBs, whose shear strength and stiffness were inferior to those of monolithic concrete bridges [10,11]. The stress in external tendons in PCSBs cannot be predicted accurately by conventional section analysis owing to strain non-compatibility of concrete and external tendons, which significantly impact the shear behavior of PCSBs with external tendons and dry joints [12]. Using high-strength concrete is an alternative consideration for enhancing the shear capacity of PCSBs with external tendons and dry joints.

Although the combination of external tendons and dry joints in precast high-strength segmental bridges (PHCSBs) can achieve accelerated construction and improved structural performance, very limited information is available on the shear behavior of this type of bridge.

The previous literature has mainly focused on the flexural behaviors of precast normal-concrete segmental beams (Ng 2003 [13]; Li 2007 [14]; Yuan et al. 2013 [15]; Jiang et al. 2016 [16]). A relatively small number of experiments have been conducted to investigate the shear behavior of precast normal-strength concrete segmental bridges.

Koseki and Breen (1983) [17] reported an experimental investigation to determine the relative shear strength across different types of joints typically used between adjacent segments of precast segmental bridges. The test results indicated substantial differences in the shear strength at a given slip in the various types of dry joints. The cylinder compressive concrete strengths of all specimens were about 50 MPa. Li (2007) [18] carried out tests on 13 monolithic beams and 14 segmental simply supported externally prestressed concrete beams and identified the differences of shear behaviors and shear strengths between them based on the existence of joints. The cubic compressive concrete strengths of all specimens were about 50 MPa. Li et al. (2013) [19], designed and conducted two series of experiments. When loading near the joint, the position of the joint had an important influence on the shear bearing capacity of the joint, and the contribution of stirrups to the shear strength of continuous beams was greater than that of simply supported beams. The prism compressive concrete strengths of specimens were less than 63.2 MPa. Yuan et al. (2015) [20] experimentally investigated the flexural and shear behaviors of segmental concrete box beams with hybrid tendons. The experimental results showed that the ratio of the number of internal tendons to the number of external tendons had a significant effect on the load-carrying capacity, ductility, and failure mode of the beams. The cubic compressive concrete strengths of specimens were 32.2 MPa. Jiang et al. (2018) [21] tested a total of 14 specimens with external tendons and dry joints, which included 3 monolithic specimens, 6 segmental specimens with dry joints, and 5 segmental specimens with epoxy joints. The experimental results demonstrated that the actual prestressing forces in external tendons could accurately predict the ultimate shear strengths of beams with external tendons at different shear span depth ratios. The cylinder compressive concrete strengths of all beams were less than 55 MPa. Jiang et al. (2019) [22] tested 9 specimens to investigate the influences of the tendon layout (hybrid tendons and external tendons), shear span ratio, and joint type (monolithic joints and dry joints) on the shear behavior of PCSBs with hybrid tendons. The test results indicated that hybrid tendons slightly improved the shear strength and stiffness of PCSBs with dry joints compared to those with external tendons. The cylinder compressive strength of concrete in this test was 36.8 MPa.

Most previous research focused on concrete specimens whose strength was lower than 65 MPa, mostly 55 MPa. Research on the application of high-strength concrete is scarce, even though high-strength concrete (HSC) is widely used in the construction industry because of its different engineering characteristics and economic advantages from normal-strength concrete. It can be used for PCSBs to reduce the dead load, save costs in bridge maintenance, prolong service life, and improve structural behavior. In addition, high-strength concrete has a high uniform density and very low impermeability, which is conducive to the durability of concrete buildings [23,24].

Few experimental results have been reported in the literature on the shear behavior of PHCSBs with compressive concrete strength exceeding 55 MPa, especially about the combined use of dry joints and external tendons. The applicability of the existing formulae of normal-strength concrete extended to the shear strength of ED-PHCSBs remains questionable [25–27]. Therefore, it would be beneficial to study the shear behavior of ED-PHCSBs and investigate the whole process, from cracking initiation to failure of ED-PHCSBs under shear loading. In this study, eight ED-PHCSBs and one externally prestressed high-strength concrete monolithic beam were designed and tested. The effect of different parameters on the shear strength of ED-PHCSBs was investigated. Experimental data of ED-PHCSBs was obtained and analyzed. The predicted values of shear strength by AASHTO 2017 and Chinese Code 2018 were compared with the experimental results. A formula for more accurately calculating the shear capacity of ED-PHCSBs based on the modified strut-and-tie model is also proposed.

## 2. Experimental Program

The experimental program included eight ED-PHCSBs and one externally prestressed high-strength concrete monolithic beam. Five parameters were considered in the research, including concrete strength, shear span-depth ratio, stirrup ratio, joint number, and joint location.

A nomenclature system and the testing parameters of the experimental beams are listed in Table 1. The specimen identifier indicates the construction method (M denotes monolithic beams and S denotes segmental beams), shear span-depth ratio (1.3, 1.8, and 2.3), target cylinder compressive strength (55 MPa, 85 MPa, and 115 MPa), stirrup ratio (P37 represents 0.37% stirrup ratio and P46 represents 0.46% stirrup ratio), joint number (N2 denotes that the joint number is 2, and N4 denotes that the joint number is 4), and joint location (b280 and b420 represent the distance from support point to first joint b1 is 280 mm and 420 mm, respectively, as shown in Figure 1). For example, S2.3-C85-P37-N4-b280 indicates a segmental beam with a 2.3 shear span-depth ratio, C85 concrete, 0.37% stirrup ratio, and 4 joints; the distance from support point to the nearest joint of this beam is 280 mm.

**Table 1.** Parameters of testing specimens.

| Specimens | Stirrup Ratio Shear Span Number (%) | Concrete Strength $f_c'$ (MPa) | Number of Joints | Shear Span Ratio (a/h) | Test Parameter |
|---|---|---|---|---|---|
| M1.3-C85-P37-None-None | 0.37 | 81.09 | 0 | 1.3 | Construction method |
| S1.3-C85-P37-N2-b280 | 0.37 | 84.30 | 2 | 1.3 | Benchmark beam |
| S1.3-C85-P46-N2-b280 | 0.46 | 84.30 | 2 | 1.3 | Stirrup ratio |
| S1.8-C85-P37-N2-b280 | 0.37 | 86.64 | 2 | 1.8 | Shear span ratio |
| S2.3-C85-P37-N2-b280 | 0.37 | 85.46 | 2 | 2.3 | Shear span ratio |
| S1.3-C115-P37-N2-b280 | 0.37 | 114.85 | 2 | 1.3 | Concrete strength |
| S1.3-C55-P37-N2-b280 | 0.37 | 53.97 | 2 | 1.3 | Concrete strength |
| S1.8-C85-P37-N2-b420 | 0.37 | 85.46 | 2 | 1.8 | Joint location |
| S2.3-C85-P37-N4-b280 | 0.37 | 83.12 | 4 | 2.3 | Joint number |

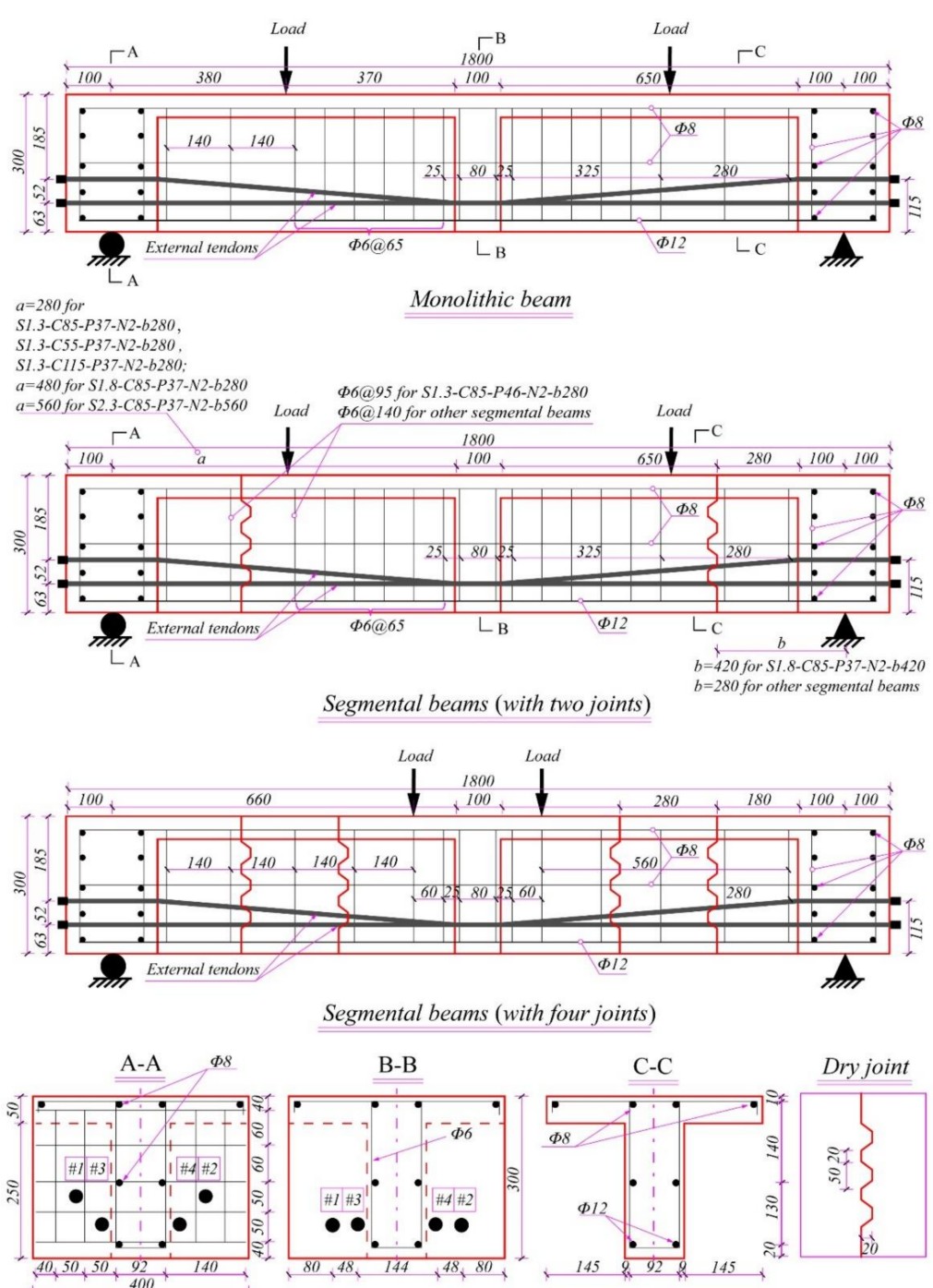

**Figure 1.** Geometric details and reinforcement arrangements of test specimens.

### 2.1. Design of Specimens

All beams had a similar profile of a T-shaped cross-section with 0.4 m width and 0.3 m height. The total length of testing beams was 1.8 m, and the clear span from one support to another was 1.6 m. The web width and the flange depth were 0.1 m and 0.05 m, respectively. Each beam was provided with a deviator 100 mm in width at the midspan. Two concrete blocks were arranged at the beam ends for the anchorage of the external tendons.

Two steel bars with a diameter of 12 mm were placed in the bottom of each beam as the longitudinal reinforcements, and several steel bars with a diameter of 8 mm were placed in the top flange as the longitudinal constructional reinforcements. For segmental beams, all longitudinal reinforcements were cut off at the joint locations, whereas the longitudinal reinforcement of the monolithic beam was continuous. The spacing of the double-leg

stirrups was 140 mm and 65 mm in the shear region and flexural region, respectively. Specifically, the stirrup spacing in the testing region of Beam S1.3-C85-P46-N2-b280 was 95 mm. Four Φ 9.50 steel strands (with a nominal diameter of 9.5 mm and a nominal section area of 54.8 mm$^2$) were selected as external prestressing tendons. In addition, a large number of reinforcing bars were used at two end blocks and one deviator of each specimen for local reinforcements. Detailed specimen dimension, reinforcement arrangement, and prestressing strand layout are shown in Figure 1.

### 2.2. Materials of Specimens

Three concrete mixtures with different strengths were used to fabricate the testing beams. The concrete mixes of C55 and C85 consisted of ordinary Portland cement (P.O 42.5), local river medium sand, graded crushed gravel (maximum size of aggregate was 12 mm), and water reducer. Their water-cement ratios were 0.49 and 0.31, respectively. The concrete mix of C115 consists of Portland cement (P·II 52.5R), fine sand, silica fume, and water reducer. The water-cement ratio of C115 was 0.22. Potable water was used for all concrete mixtures. When fabricating the specimens, an adequate number of standard cylinder specimens (150 mm in diameter and 300 mm in height) were cast to measure the material properties and cured in an outdoor ambient environment before testing. Compressive strength, modulus of elasticity, and Poisson's ratio of each concrete mixture were tested according to (American Society for Testing and Materials) ASTM [28]. Mixture proportions and other properties of the concrete mixtures are summarized in Tables 2 and 3.

**Table 2.** Properties of concrete.

| Concrete Type | Concrete Strength (MPa) | | | | | Elastic Modulus (GPa) | | | | |
|---|---|---|---|---|---|---|---|---|---|---|
| | 1 | 2 | 3 | Average Value (MPa) | Standard Deviation | 1 | 2 | 3 | Average Value (GPa) | Standard Deviation |
| C55 | 53.97 | 57.41 | 50.53 | 53.97 | 2.81 | 32.30 | 40.32 | 35.41 | 36.01 | 3.30 |
| C85 | 82.73 | 88.13 | 83.78 | 84.88 | 2.34 | 40.80 | 40.18 | 39.59 | 40.19 | 2.21 |
| C115 | 114.86 | 116.04 | 113.65 | 114.85 | 0.98 | 44.56 | 48.48 | 43.28 | 45.44 | 0.49 |

Note: Three characterization specimens of the tested beams were tested to obtain the mean value of the concrete.

**Table 3.** Design mix proportions of concrete.

| Concrete Type | Mixture Quantity (kg/m$^3$) | | | | | |
|---|---|---|---|---|---|---|
| | Coarse Aggregate | Sand | Water | Cement | Silica Fume | Super-Plasticizer |
| C55 | 1234 | 608 | 200 | 408 | — | 4.88 |
| C85 | 1208 | 650 | 140 | 447 | — | 9.76 |
| C115 | — | 960 | 232 | 800 | 240 | 46 |

Note: "—" represented the concrete type without adding the materials.

The yield strength and elastic modulus of the conventional reinforcements were tested as 454.4 MPa and $2.013 \times 10^5$ MPa, respectively. The yield strength and elastic modulus of prestressing strands provided by the manufacturer were 1815 MPa and $1.932 \times 10^5$ MPa, respectively.

### 2.3. Fabrication of Specimens

All the segmental beams were fabricated by long-line match casting technology. Before casting concrete, prestressing tendon ducts were placed at designated specific positions to arrange the external tendons. To obtain the shape of dry joints, the key-shaped steel formwork was fixed in the designated specific positions of the wood formwork. For the three-part segmental beams, the middle segment was cast first. After 24 h curing, the key-shaped steel formwork was unmolded, and side segments were subsequently cast. The wood formwork of beams was unmolded after 7-day curing in an outdoor ambient environment.

For the five-part segmental beams, only odd-numbered segments of ED-PHCSBs were cast. Then, the wood module and steel formworks of the shear key were dismantled after 3 days when the concrete had finally been set. Lastly, the remaining segments were fabricated to match the existing segments. Monolithic beams were cast at one time. The fabricating process of the segmental beams and monolithic beams is shown in Figure 2.

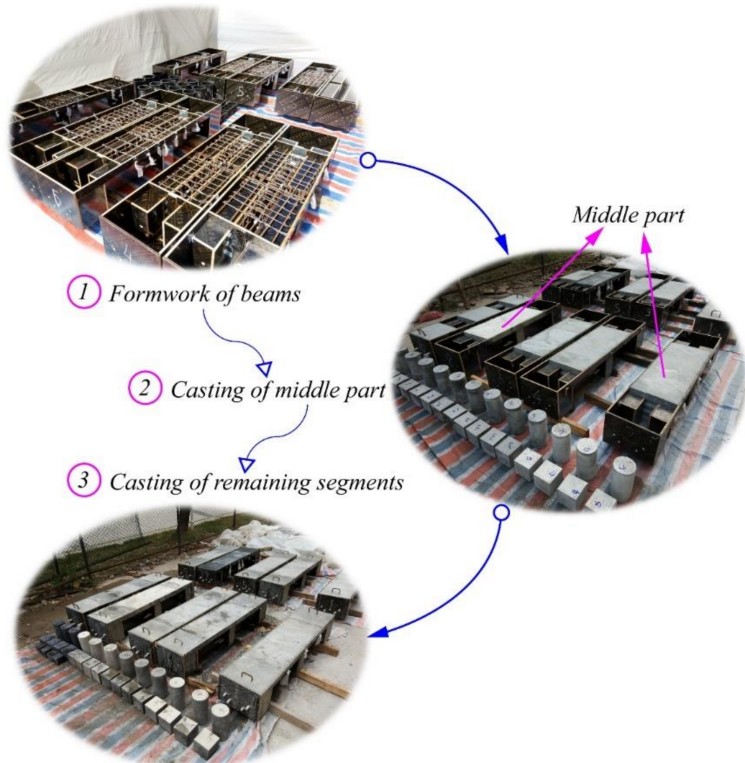

**Figure 2.** Casting procedure of test beams.

Table 4 shows the prestressing details of all specimens. After natural curing for more than 28 days, the segments were assembled, and then the prestressing force of external tendons was applied. Each tendon was tensioned to 0.40 to 0.60 $f_u$ ($f_u$ is the tensile strength standard value of tendons, $f_u$ = 1860 MPa). For inevitable prestress losses, the magnitude of effective prestresses ($f_{pe}$) in all specimens was later found to be in the range of 0.35 to 0.55 $f_u$.

**Table 4.** Summary of test results.

| Specimens | Ultimate Loads (kN) | Cracking Loads (kN) | Stress Increments of External Tendon $\Delta f$ (MPa) | | | | Maximum Deflection at Midspan (mm) | Failure Modes |
|---|---|---|---|---|---|---|---|---|
| | | | #1 | #2 | #3 | #4 | | |
| M1.3-C85-P37-None-None | 729.0 | 364.6 | 771.9 | 817.5 | 927.0 | 470.8 | 23.604 | SC |
| S1.3-C85-P37-N2-b280 | 487.5 | 247.9 | 729.9 | 746.4 | 875.9 | 890.5 | 19.228 | SC |
| S1.3-C85-P46-N2-b280 | 565.2 | 253.1 | 618.6 | 715.3 | 833.9 | 885.0 | 12.646 | SC |
| S1.8-C85-P37-N2-b280 | 360.1 | 254.3 | 372.3 | 344.9 | 476.3 | 465.3 | 9.828 | SC |
| S2.3-C85-P37-N2-b280 | 358.0 | 218 | 788.3 | 801.1 | 861.3 | 852.2 | 20.934 | SC |
| S1.3-C115-P37-N2-b280 | 556.0 | 230 | 593.1 | 573 | 709.9 | 691.6 | 16.636 | AT |
| S1.3-C55-P37-N2-b280 | 363.0 | 222.9 | 386.9 | 390.5 | 498.2 | 505.5 | 7.937 | SC |
| S1.8-C85-P37-N2-b420 | 402.2 | 180 | 625.9 | 636.9 | 788.3 | 768.2 | 18.746 | SC |
| S2.3-C85-P37-N4-b280 | 339.5 | 156.8 | 660.6 | 748.2 | 863.1 | 826.6 | 23.681 | SC |

Note: SC = shear compression, and the AT = abruption of the external tendon.

### 2.4. Test Setup and Instrumentation

The simply supported beams were tested under four-point loading as illustrated in Figure 3, which was arranged by roller support at one terminal and hinge support at another. The vertical load was provided by a jack with a capacity of 1000 kN, model YCW100B20 (manufactured by OVM Machinery Co., Ltd., Liuzhou, China). A spandrel beam with adjustable length was used to transfer vertical load from the jack to the beam at two specific positions.

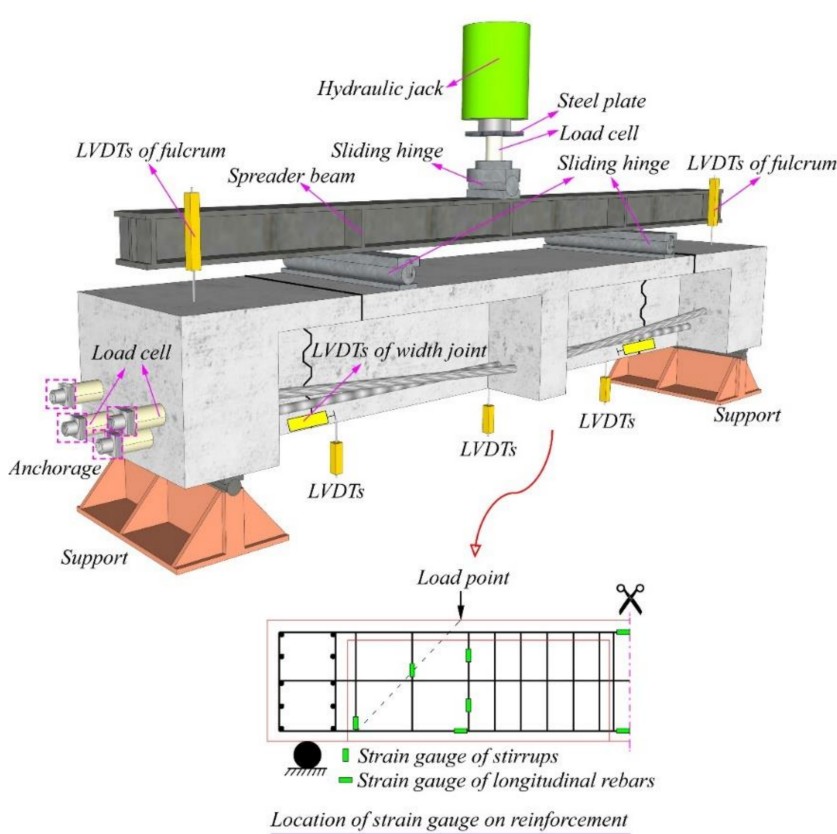

**Figure 3.** Typical experimental setup for test specimens.

The previous record showed that the maximum deflection at failure was always measured at the loading points or the midspan of testing beams [16]. Thus, five linear variable displacement transducers (LVDTs) measuring the beam deflections were installed at midspan, loading points, and support points. Moreover, an LVDT with a range of 10 mm was horizontally installed at the bottom of each joint to monitor the open width of joints. The vertical force was measured with a load cell under the jack, and tension forces of prestressing tendons were supervised with four hollow load cells which deployed at the end of each tendon. The locations of instruments are also shown in Figure 3.

In addition, steel strain gauges were glued at the intersection point of the selected stirrups and the lines connecting the loading point and the support. The strain at the mid-span of longitudinal reinforcements was also measured. The detailed locations of these strain gauges are illustrated in Figure 3.

A 10 kN preload was applied to ensure that all testing facilities worked normally. During formal testing, the loading was applied monotonically every 5 min at an increment of 20 kN before cracking and then at an increment of 10 kN until failure. After each loading step, the initiating and propagating of cracks were observed and marked. Throughout the test process, the applied load force, the forces of the prestressing tendons, the deflections of the beam at five measuring points, the widths of joint openings, and all strain gauges were collected.

## 3. Test Results and Discussion

### 3.1. Cracking Propagations and Failure Modes

Cracking propagations and failure modes of all beams are exhibited in Figure 4. For the monolithic beam of M1.3-C85-P37-N2-b280, after a vertical load of about 346 kN (47.5% $P_u$, where $P_u$ was the ultimate load of the test beams), several flexural cracks appeared, initially at the bottom of the beam below the loading point. When the load increased, more flexural cracks appeared at different positions at the bottom of the beam and propagated upward vertically. When the loading reached 364.6 kN (50% $P_u$), diagonal shear cracks appeared on the web in the shear zone and extended toward the loading point and supporting point. Finally, when the loading reached its ultimate load $P_u$ (729.0 kN), both flexural and shear cracks expanded to connect the bottom of the beam web and the flange.

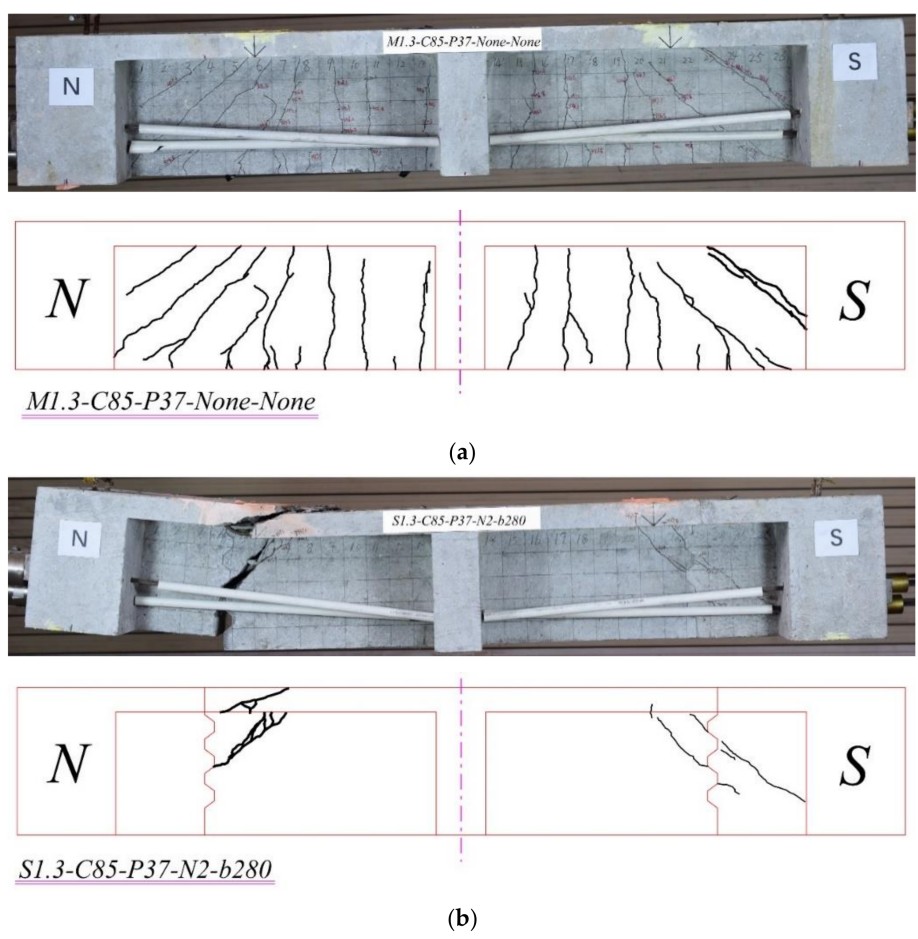

(a)

(b)

**Figure 4.** *Cont.*

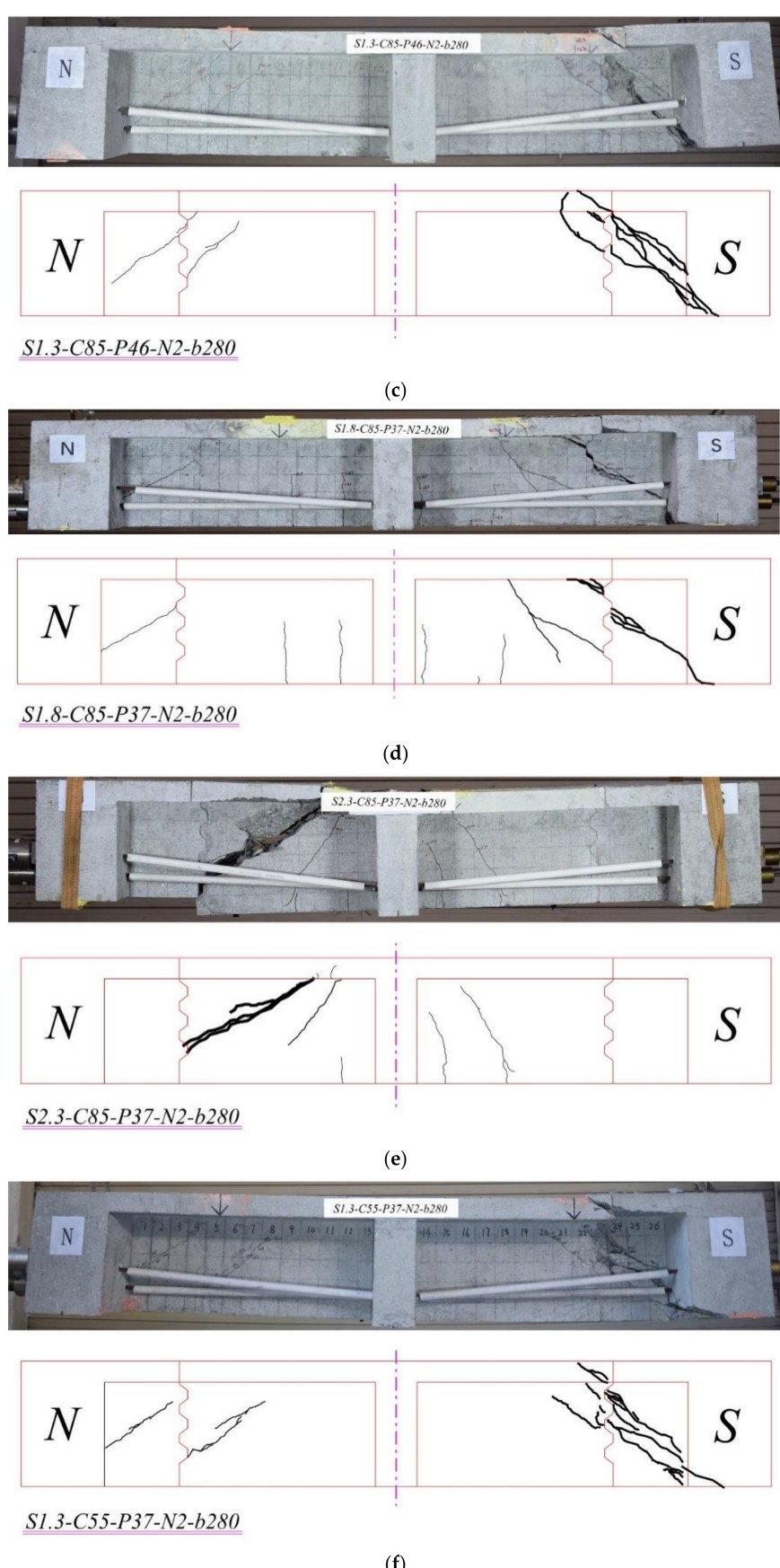

(c)

(d)

(e)

(f)

**Figure 4.** *Cont.*

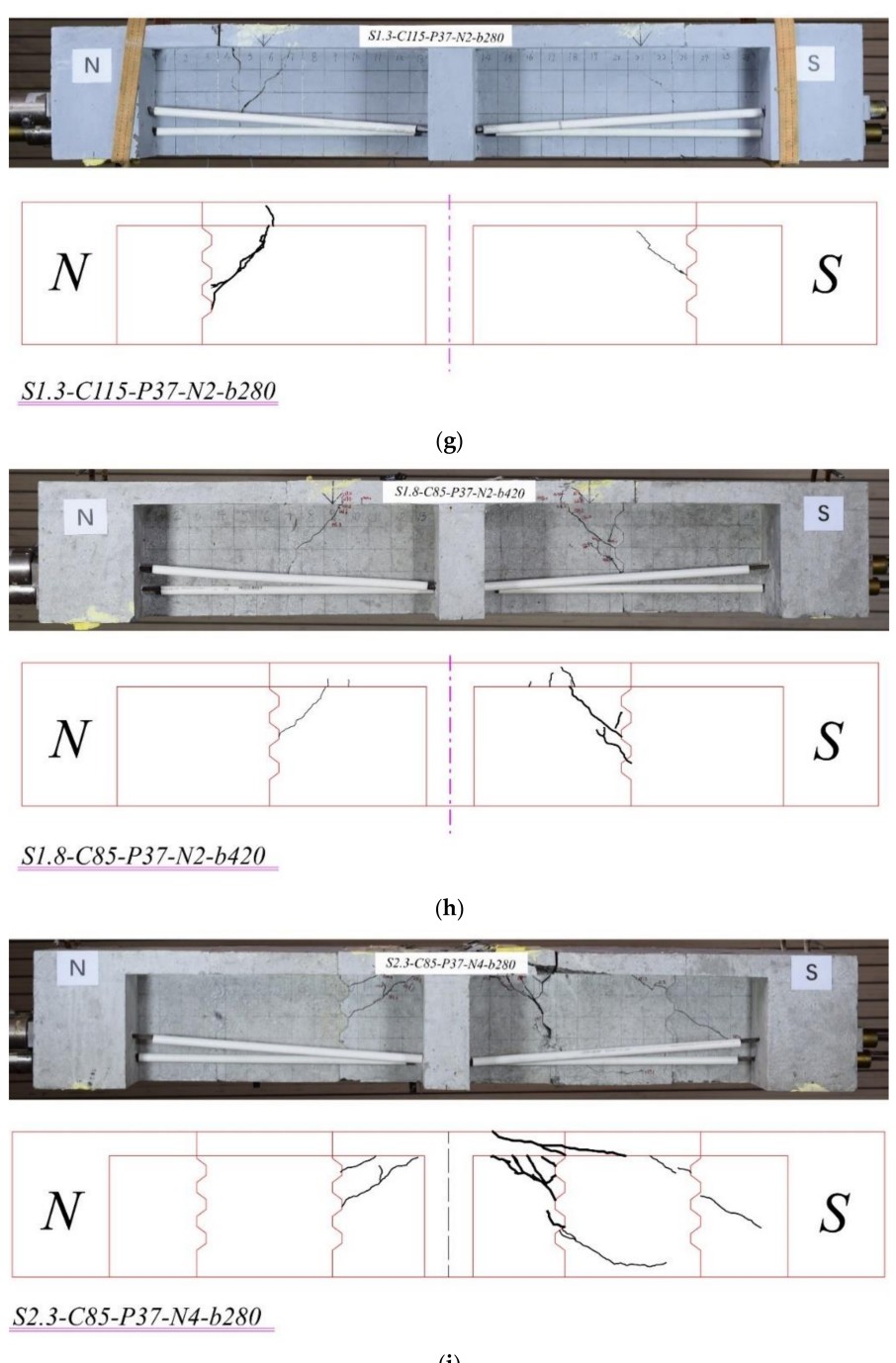

**Figure 4.** Depictions of failure modes of all the test beams. (**a**) M1.3-C85-P37-None-None. (**b**) S1.3-C85-P37-N2-b280. (**c**) S1.3-C85-P46-N2-b280. (**d**) S1.8-C85-P37-N2-b280. (**e**) S2.3-C85-P37-N2-b280. (**f**) S1.3-C55-P37-N2-b280. (**g**) S1.3-C115-P37-N2-b280. (**h**) S1.8-C85-P37-N2-b420. (**i**) S2.3-C85-P37-N4-b280.

For the segmental beams, there were barely diagonal cracks on the webs. The initial shear cracks of segmental beams generally appeared at the roots of the shear keys because of the stress concentration. As the vertical load increased, the cracks developed gradually along the line connecting the support and the loading point. In most cases, one of these cracks developed as a primary crack which would divide the beam into two parts and cause the failure of the beam. The failure mode, in this case, was described as shear compression failure (SC), e.g., S1.3-C85-P37-N2-b280, S1.3-C85-P46-N2-b280, S1.8-C85-P37-N2-b280, S2.3-C85-P37-N2-b280, S1.3-C55-P37-N2-b280, S1.8-C85-P37-N2-b420, and S2.3-C85-P37-

N4-b280. In another case, the prestressing tendons broke before concrete failure, such as S1.3-C115-P37-N2-b280. The failure mode, in this case, was described as an abruption of the external tendons (AT). For Beams S1.8-C85-P37-N2-b280 and S2.3-C85-P37-N2-b280, there were several tiny flexural cracks at the bottom of the webs; however, the influence of these flexural cracks on the mechanical shear performance of the segmental beams was negligible.

Comparing cracking propagations among beams with different strength concretes, for beams with the higher strength concrete, fewer cracks were observed on the surfaces of precast concrete segmental beams when the failure occurred, as shown in Figure 4.

Beam S1.3-C85-P46-N2-b280 (95 mm stirrup spacing) was similar to Beam S1.3-C85-P37-N2-b280 (140 mm stirrup spacing) except for the stirrup ratio. Experimental results showed that more shear cracks appeared in Beam S1.3-C85-P46-N2-b280 with a larger stirrup ratio. This phenomenon indicated that shear stirrups played an important role in the crack distributions.

### 3.2. Deflection Characteristics

Figure 5 shows the deflections along the beam length for each beam at different vertical loads. Compared with the reference beams of M1.3-C85-P37-None-None, the mid-span displacement of S1.3-C85-P37-N2-b280 was significantly larger than M1.3-C85-P37-None-None when the vertical load reached the ultimate load. It was indicated that the stiffness of the monolithic beams was larger than the segmental beams. This conclusion was also verified by Jiang [21]. In addition, the displacement of all the test beams was symmetrical along the mid-span before the vertical load reached the 0.4 Vu (where Vu represented the ultimate load of beams). After that, the asymmetrical deformation along the mid-span was observed in the test. The reason was that one side of the beam cracked earlier than the other, which resulted in the deflection at this side being larger than on the other side near the vertical ultimate load Vu.

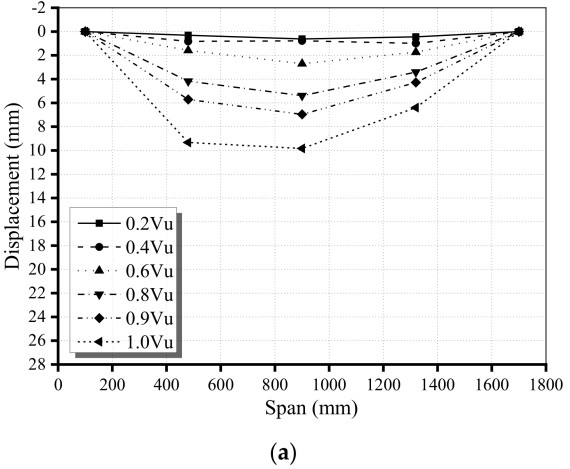

(a)

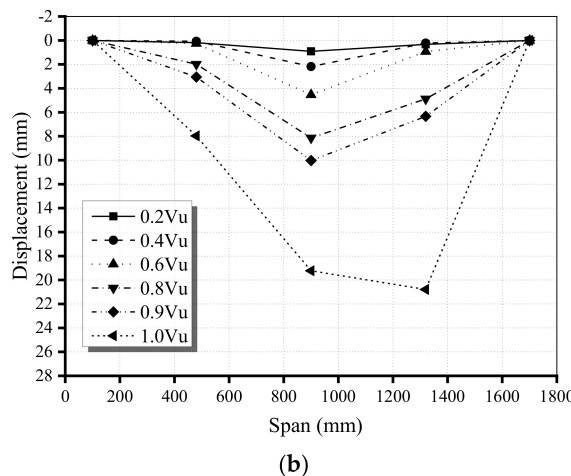

(b)

**Figure 5.** *Cont.*

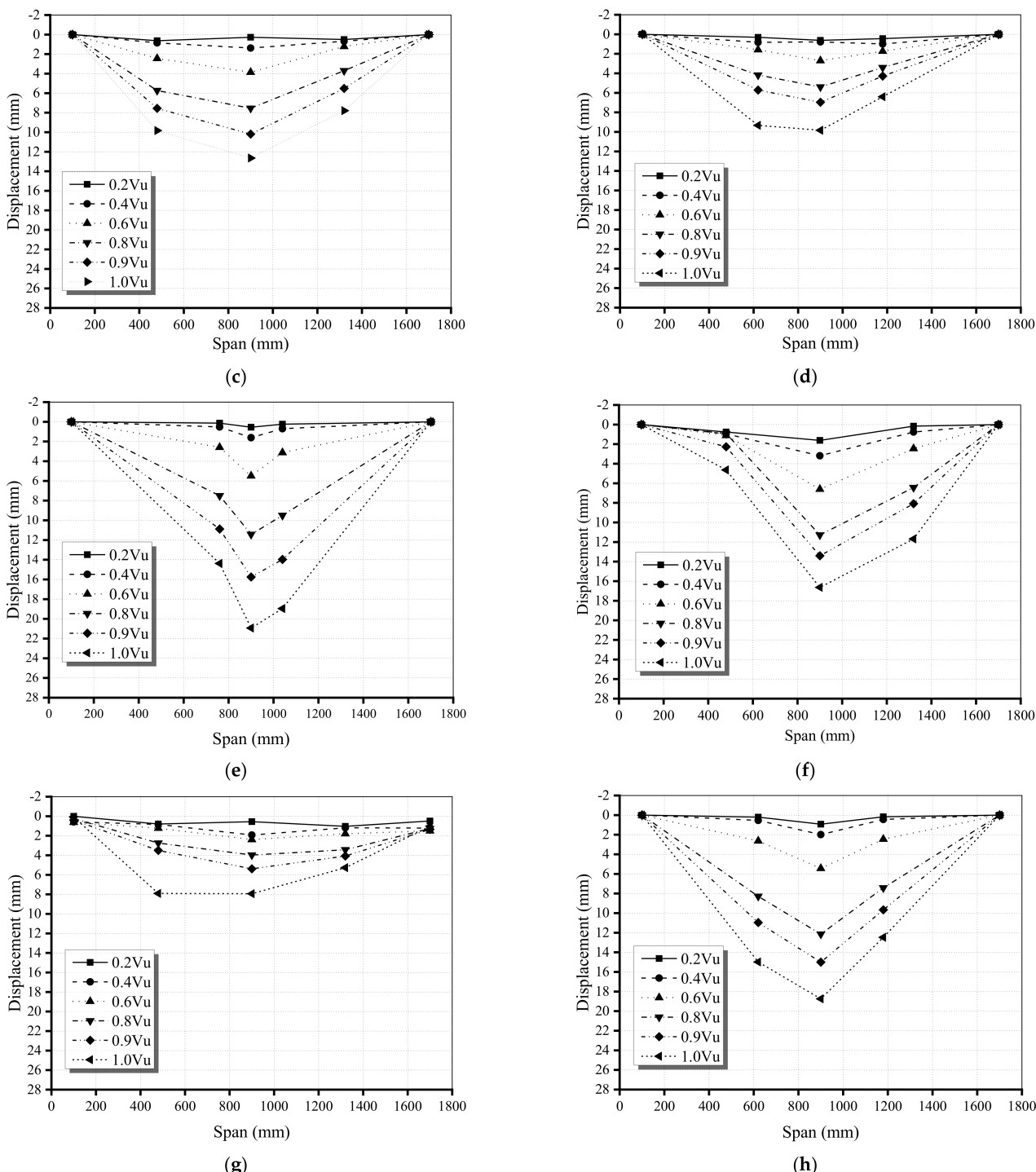

**Figure 5.** *Cont.*

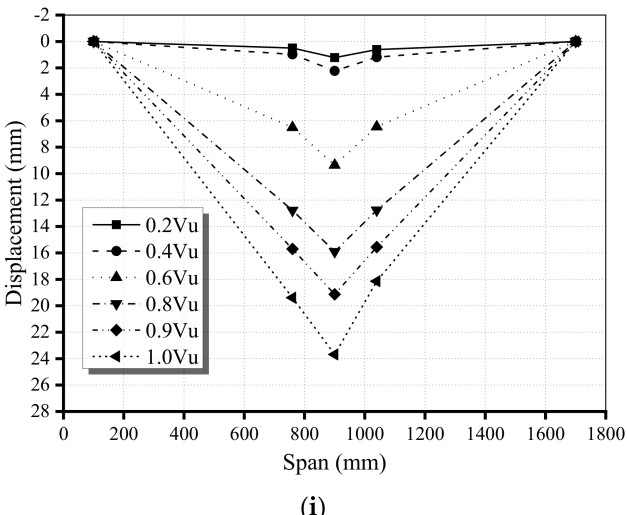

(**i**)

**Figure 5.** Displacements along beam span under load. (**a**) M1.3-C85-P37-None-None. (**b**) S1.3-C85-P37-N2-b280. (**c**) S1.3-C85-P46-N2-b280. (**d**) S1.8-C85-P37-N2-b280. (**e**) S2.3-C85-P37-N2-b280. (**f**) S1.3-C115-P37-N2-b280. (**g**) S1.3-C55-P37-N2-b280. (**h**) S1.8-C85-P37-N2-b420. (**i**) S2.3-C85-P37-N4-b280.

Figure 6 shows the load-displacement at mid-span curves for all beams. At first, the deflection increased linearly, but starting from the specified point of the cracking load, this response became nonlinear. The stiffness of the beams also reduced after cracking.

By comparing the load-displacement curves at the mid-span of each beam, the influences of experimental parameters on the load-displacement relationships of the beams can be revealed. Figure 6a shows the load-displacement curves of beams with different strength concrete. The slope of the curves on the specimens of S1.3-C55-P37-b280, S1.3-C85-P37-b280, and S1.3-C115-P37-b280 was basely comparable before cracking. However, for the S1.3-C55-P37-b280, the slope of the curve descended rapidly after cracking. For the high-strength concrete, such as S1.3-C85-P37-b280 and S1.3-C115-P37-b280, the slopes of the curves descended slowly until they reached the ultimate loads.

As shown in Figure 6b, the slopes of the specimens (S1.3-C85-P37-b280, S1.8-C85-P37-b280, and S2.3-C85-P37-b280) were similar before cracking, but the slope of the curves of the S1.3-C85-P37-b280 was higher than that of S1.8-C85-P37-b280 and S2.3-C85-P37-b280 after cracking. It was indicated that the stiffness of the specimen decreased obviously after cracking with the increase in the shear span-depth ratio. Comparing Beam S1.3-C85-P37-N2-b280 and S1.3-C85-P46-N2-b280 in Figure 6c, it was confirmed that increasing the stirrup ratio can effectively improve the stiffness of beams. In addition, it could be inferred from Figure 6d,e, the stiffness of the monolithic beam was higher than the segmental one and when the number of joints in the segmental beams increased (e.g., S2.3-C85-P37-N2-b280 and S2.3-C85-P37-N4-b280) or the joint located closer to the load point (e.g., S1.8-C85-P37-N2-b280 and S1.8-C85-P37-N2-b420), the stiffness reduced and the deflection enhanced.

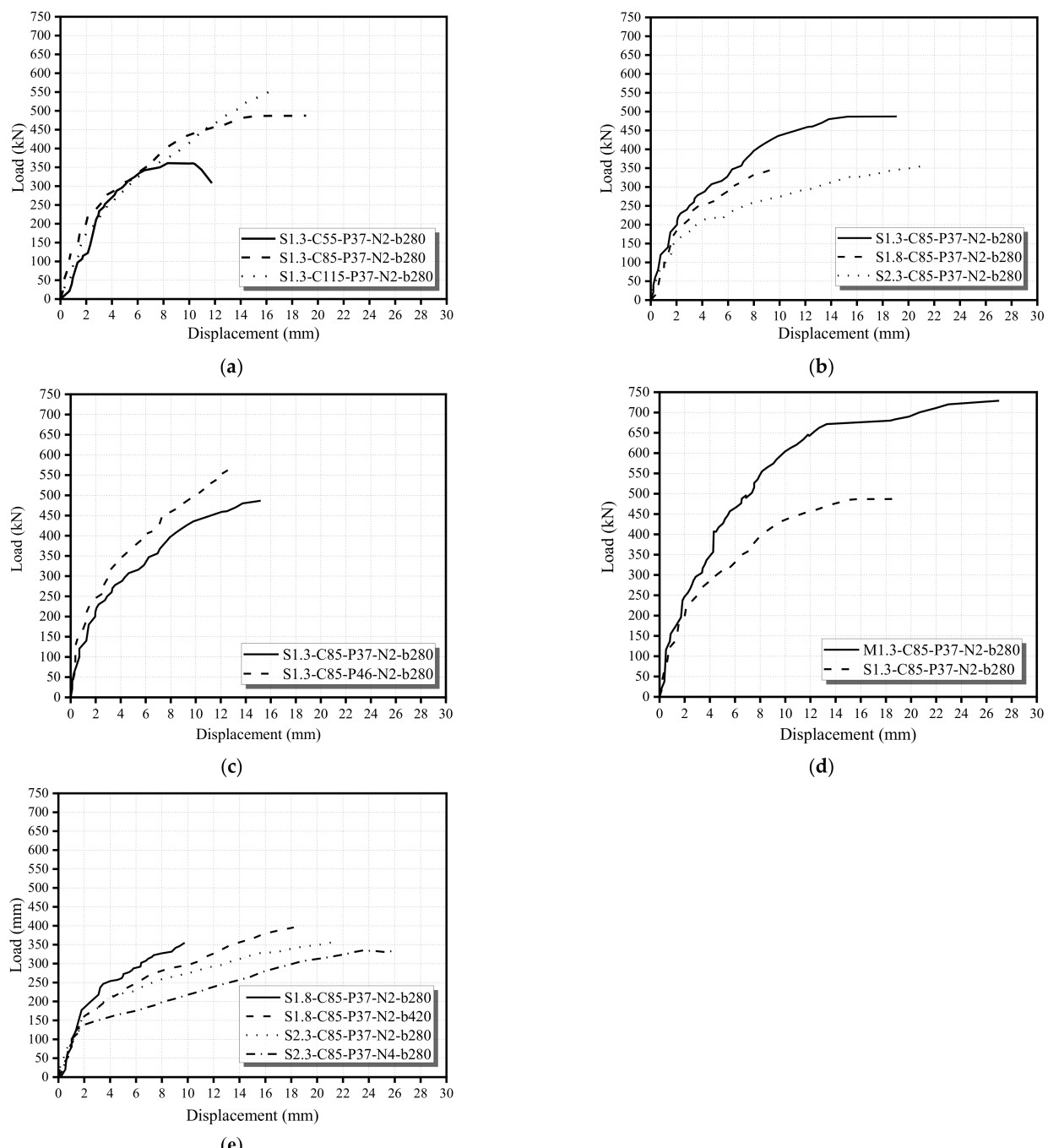

**Figure 6.** Load-displacement of curves. (**a**) strength of concrete group. (**b**) shear-span-depth ratio group. (**c**) stirrups ratio group. (**d**) segmental and monolithic beams group. (**e**) location of joints and joints number group.

### 3.3. Variations of Tendons Stresses

Figure 7a,b illustrates the stress variations in the straight tendons and draped tendons of specimens with different concrete strengths under vertical loads. At the beginning stage, the stress of each tendon kept approximately constant until the cracking load. When a further load was applied, with more deformation of the beams, the stresses of tendons began to increase quickly. Finally, the beams failed because of concrete crushing or tendon fracture. No matter which failure mode the beams encountered, the tendon stresses did not reach the standard tension strength of 1860 MPa, because in some cases, the concrete was crushed before the tendon stresses reached the tension strength, and in

the other case, the tendons fractured prematurely due to the shear action on prestressing tendons at the deviator.

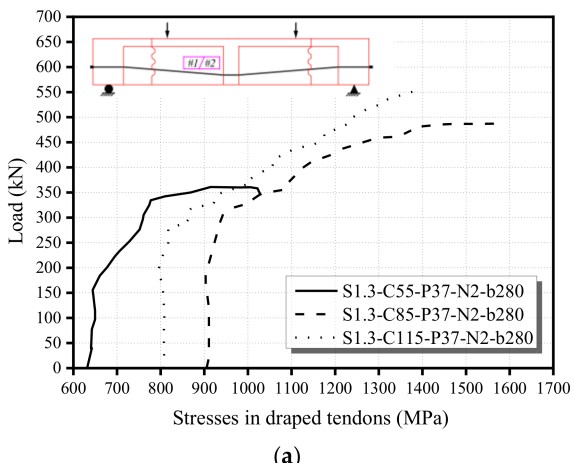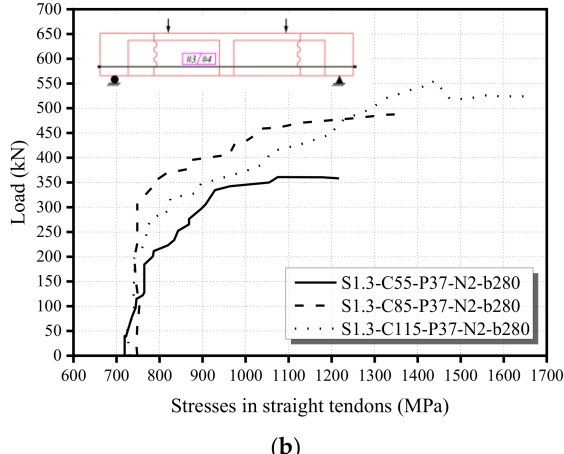

(**a**)　　　　　　　　　　　　　　　　　　　　(**b**)

**Figure 7.** Load-stress curves of draped or straight external tendons curves. (**a**) stresses in draped tendons. (**b**) stresses in straight tendons.

### 3.4. Opening Width of Joints

To analyze the behavior of joint opening, the opening width of joints on the failure side of the test beams was recorded. The opening width of the joint versus vertical loads is shown in Figure 8. Before joint opening, the opening widths of joints were kept stable within the scope of 1.0 mm. Then, the opening width almost extended linearly, with the load increasing after the opening of the joints. Finally, the curves performed more flatly until the beams approached failure, which means the opening width of joints increased continuously while shear loads were slowly increasing.

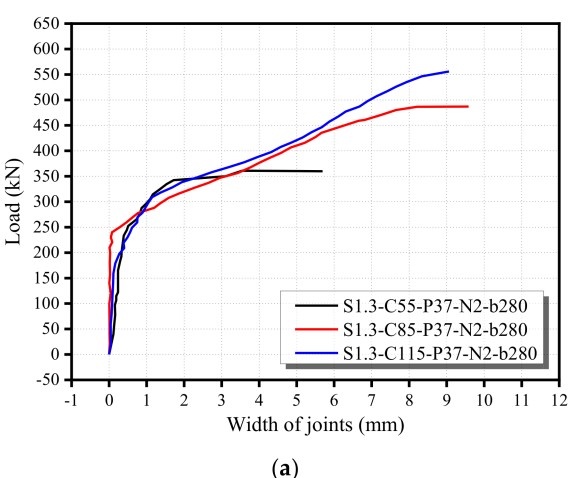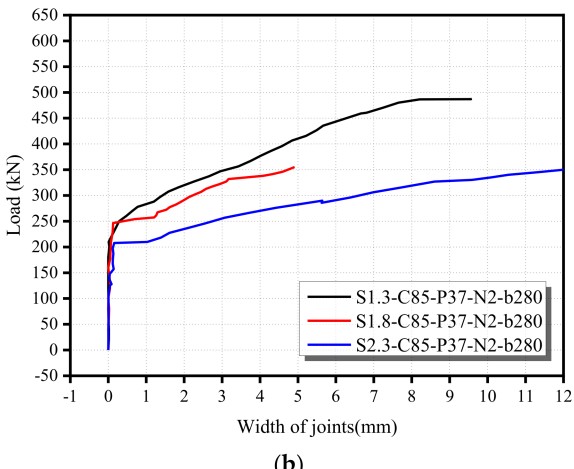

(**a**)　　　　　　　　　　　　　　　　　　　　(**b**)

**Figure 8.** *Cont.*

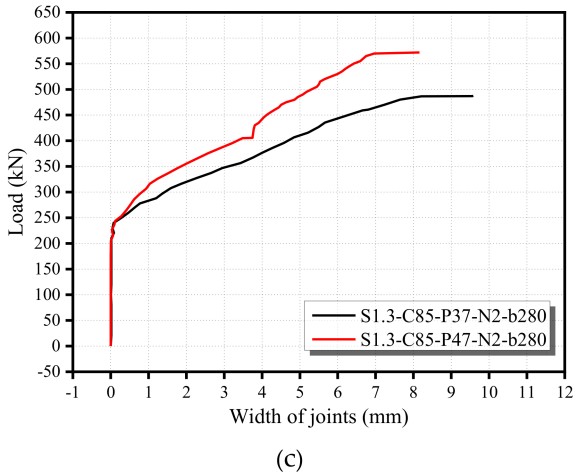

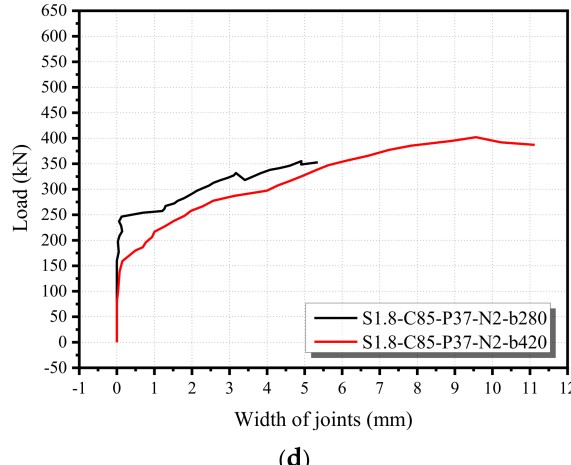

(c)

(d)

**Figure 8.** Load-joint width curves. (**a**) strength of concrete group. (**b**) shear-depth-span ratio group. (**c**) stirrups ratio group. (**d**) location of joints group.

### 3.5. Steel Strain Variations

Observing the stirrup strains in the shear zone, the stirrups were almost ineffective for shear resistance before concrete cracking. After cracking, the stirrup strains in the stirrups cross the principal crack increased with the increasing load. In general, stirrup strains in the non-cracked area were very small. Interestingly, several stirrups were compressed, but they were suddenly tensioned at concrete crushing. The load-strain curves of the stirrups are shown in Figure 9.

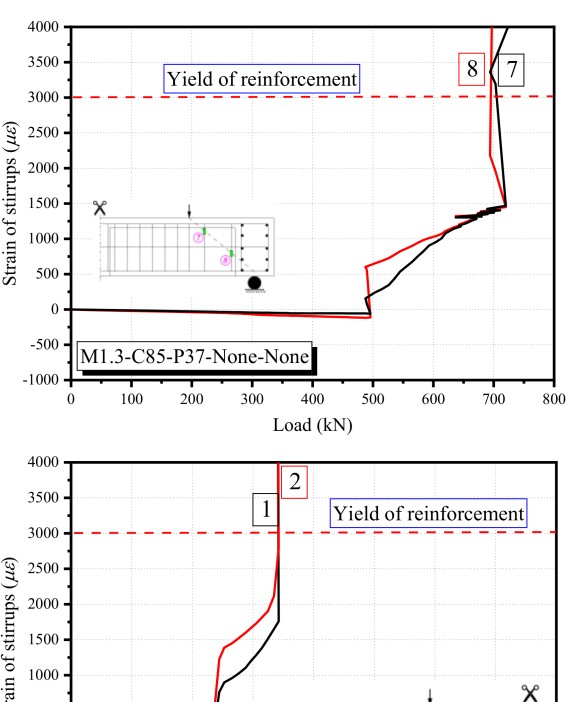

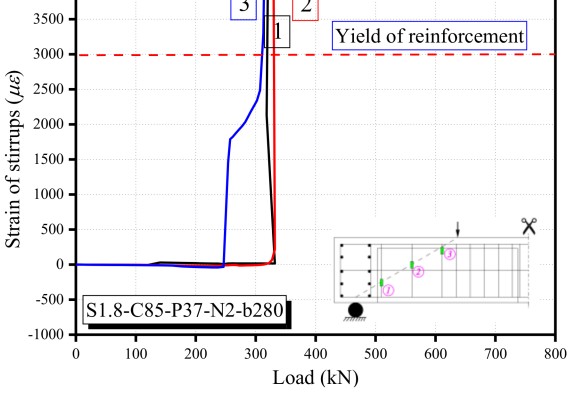

**Figure 9.** *Cont.*

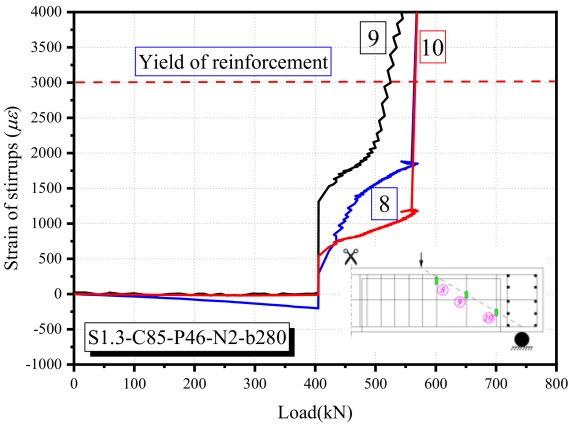
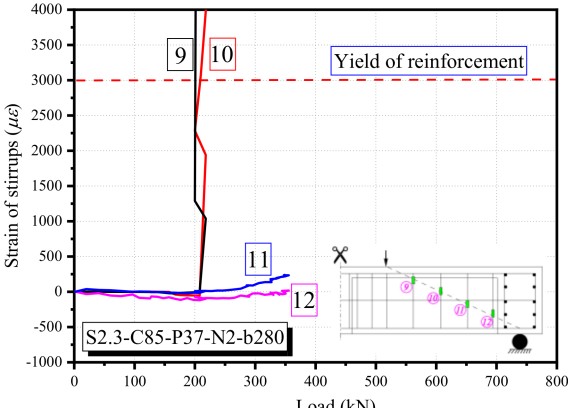

**Figure 9.** Load-stirrup strain curves.

Figure 10 shows the load-strain curves of the longitudinal reinforcements at the bottom of beams. It can be observed that the strain of the longitudinal reinforcement increased with the increasing of vertical loads in the monolithic beam. Furthermore, the strains of longitudinal reinforcement at failure were more than 0.003, which is the yield of reinforcement.

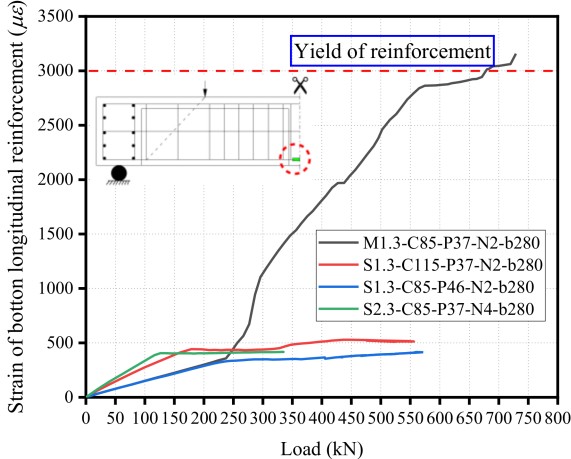

**Figure 10.** Shear capacity of all specimens.

For segmental beams, the strain of the longitudinal reinforcement behaved differently from that of the monolithic one. According to Figure 10, the strain was increased with the increase in load, but the strain of the longitudinal reinforcement of segmental beams was increased until the dry joints opening, then kept stable at a relatively low level, the strains were less than $600\ \mu\varepsilon$. The value of strain remained unchanged until the failure of the beams.

With the load increases, the longitudinal reinforcement of the monolithic beam was bearing the flexural stress. Thus, the strain of the longitudinal reinforcement increased until it reached the yield stress. Differently, the longitudinal reinforcements of segmental beams were cut at the location of the dry joints. The longitudinal reinforcements were not bearing the stress anymore; the external tendons were bearing the stress, which reflected the increase in the strain of external tendons.

### 3.6. Effect of Test Parameters on Ultimate Load Capacity

Figure 11 shows the ultimate shear capacity of all beams. It can be seen that the ultimate shear capacity of all segmental beams was lower than that of the monolithic beam.

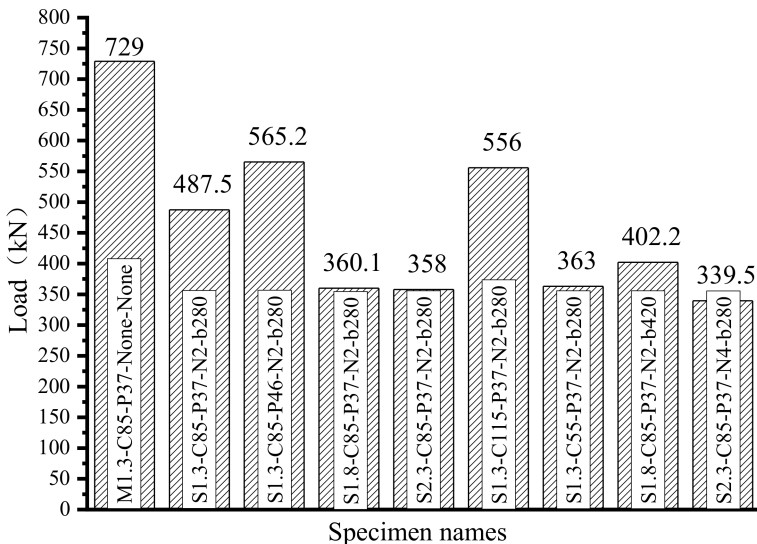

**Figure 11.** Load-strain curves of longitudinal reinforcements.

### 3.6.1. Concrete Strength

The concrete strength can largely affect the ultimate shear capacity of precast concrete segmental beams. The segmental beam with C115 concrete (S1.3-C115-P37-N2-b280) recorded the highest value of 556 kN, compared with the other two counterparts, which were 53.2% and 14.0% higher than those specimens with C55 and C85, respectively.

### 3.6.2. Shear Span-Depth Ratio

By comparing the beams with different shear span-depth ratios, it could be concluded that reducing the shear span-depth ratio would result in the higher ultimate shear capacity of beams. The segmental beam with a shear span-depth ratio of 1.3 had approximately 35.4% and 43.6% higher ultimate shear capacity than those corresponding beams with a shear span-depth ratio of 1.8 and 2.3, respectively.

### 3.6.3. Stirrup Ratio

Increasing the stirrup ratio could also enhance the ultimate shear capacity of the precast beams. The ultimate shear capacity of S1.3-C85-P46-N2-b280 was 16.0% higher than that of S1.3-C85-P37-N2-b280.

### 3.6.4. Joint Number and Joint Location

The segmental beams of S2.3-C85-P37-N2-b280 and S2.3-C85-P37-N4-b280, which possessed different joint numbers, were almost similar in ultimate shear capacity, and their deviations were within 5%. Therefore, it can be considered that joint number has little significant effect on the ultimate shear bearing capacity of ED-PHCSBs. Comparing S1.8-C85-P37-N2-b280 with S1.8-C85-P37-N2-b420, it can be inferred that the joint location had an influence on the shear behavior of ED-PHCSBs. However, the failure mechanism needs to be further investigated.

## 4. Comparison of Experimental Results with Theoretical Prediction

The shear strength from experiments was compared with the predictions from AASHTO 2017 [26] and Chinese Code 2018 [27]. Tables 5 and 6 show the experimental results and the predictions of AASHTO LRFD 2017 and Chinese Code 2018. All design safety factors were taken as 1.0 when using these specifications.

**Table 5.** Comparison of testing shear strength with calculated value of AASHTO.

| Specimens | $f_c'$ (MPa) | bv (mm) | dv (mm) | $A_v$ (mm) | θ (°) | β | s (mm) | Shear Strength (kN) | | $V_E/V_A$ |
|---|---|---|---|---|---|---|---|---|---|---|
| | | | | | | | | $V_E$ | $V_A$ | |
| M1.3-C85-P37-None-None | 81.09 | 110.00 | 280.00 | 57.00 | 45.00 | 2.00 | 140 | 364.50 | 106.45 | 3.42 |
| S1.3-C85-P37-N2-b280 | 84.30 | 110.00 | 280.00 | 57.00 | 45.00 | 2.00 | 140 | 243.75 | 107.37 | 2.27 |
| S1.3-C85-P46-N2-b280 | 84.30 | 110.00 | 280.00 | 57.00 | 45.00 | 2.00 | 95 | 286.00 | 120.11 | 2.38 |
| S1.8-C85-P37-N2-b280 | 86.64 | 110.00 | 280.00 | 57.00 | 45.00 | 2.00 | 140 | 180.05 | 108.02 | 1.67 |
| S2.3-C85-P37-N2-b280 | 85.46 | 110.00 | 280.00 | 57.00 | 45.00 | 2.00 | 140 | 179.00 | 107.69 | 1.66 |
| S1.3-C115-P37-N2-b280 | 114.8 | 110.00 | 280.00 | 57.00 | 45.00 | 2.00 | 140 | 278.00 | 115.32 | 2.41 |
| S1.3-C55-P37-N2-b280 | 53.97 | 110.00 | 280.00 | 57.00 | 45.00 | 2.00 | 140 | 181.50 | 97.86 | 1.85 |
| S1.8-C85-P37-N2-b420 | 85.46 | 110.00 | 280.00 | 57.00 | 45.00 | 2.00 | 140 | 201.10 | 107.69 | 1.87 |
| S2.3-C85-P37-N4-b280 | 83.12 | 110.00 | 280.00 | 57.00 | 45.00 | 2.00 | 140 | 169.50 | 107.03 | 1.58 |

For all beams: the average of $V_E/V_A$ is 2.12, with the standard deviations of 0.58.

Note: 1 in. = 25.4 mm, 1000 psi = 6.895 MPa, 1 lb = 4.45 N; $V_E$ is the experimental value, and the $V_A$ is the calculated value.

**Table 6.** Comparison of testing shear strength with calculated value of Chinese Code.

| Specimens | $f_{cu,k}'$ (MPa) | b (mm) | $h_0$ (mm) | $A_{ex}$ (mm$^2$) | $\theta_{ex}$ (°) | P | Shear Strength (kN) | | $V_E/V_J$ |
|---|---|---|---|---|---|---|---|---|---|
| | | | | | | | $V_E$ | $V_J$ | |
| M1.3-C85-P37-None-None | 81.09 | 110.00 | 280.00 | 282.99 | 5.00 | 1.65 | 364.50 | 134.98 | 2.70 |
| S1.3-C85-P37-N2-b280 | 84.30 | 110.00 | 280.00 | 282.99 | 5s.00 | 0.92 | 243.75 | 126.02 | 1.93 |
| S1.3-C85-P46-N2-b280 | 84.30 | 110.00 | 280.00 | 282.99 | 5.00 | 0.92 | 286.00 | 139.58 | 2.05 |
| S1.8-C85-P37-N2-b280 | 86.64 | 110.00 | 280.00 | 282.99 | 5.00 | 0.92 | 180.05 | 126.20 | 1.43 |
| S2.3-C85-P37-N2-b280 | 85.46 | 110.00 | 280.00 | 282.99 | 5.00 | 0.92 | 179.00 | 126.10 | 1.42 |
| S1.3-C115-P37-N2-b280 | 114.85 | 110.00 | 280.00 | 282.99 | 5.00 | 0.92 | 278.00 | 135.04 | 2.06 |
| S1.3-C55-P37-N2-b280 | 53.97 | 110.00 | 280.00 | 282.99 | 5.00 | 0.92 | 181.50 | 111.51 | 1.63 |
| S1.8-C85-P37-N2-b420 | 85.46 | 110.00 | 280.00 | 282.99 | 5.00 | 0.92 | 201.10 | 126.22 | 1.59 |
| S2.3-C85-P37-N4-b280 | 83.12 | 110.00 | 280.00 | 282.99 | 5.00 | 0.92 | 169.50 | 125.83 | 1.35 |

For all beams: the average of $V_E/V_A$ is 1.80, with the standard deviations of 0.44.

Note: Coefficient of $\alpha_1$, $\alpha_2$, and $\alpha_3$ is 1.0, 1.25, and 1.1, respectively. $V_E$ is the experimental value, and the $V_J$ is the calculated value.

*4.1. AASHTO 2017*

The nominal shear resistance $V_n$, should be determined as the minimum of the following:

$$V_n = V_c + V_s + V_p \tag{1}$$

where $V_c$ is the shear contribution of concrete (kN); $V_s$ is the shear contribution of stirrups.

$$V_n = 0.25 f_c' b_v d_v + V_p \tag{2}$$

In which:

$$V_c = 0.0316 \beta \sqrt{f_c'} b_v d_v \tag{3}$$

$$V_s = \frac{A_v f_y d_v (\cot\theta + \cot\alpha) \sin\alpha}{s} \tag{4}$$

where $\theta$ is the angle of inclination of diagonal compressive stresses ($\theta = 45°$ herein); $d_v$ is effective shear depth; $s$ is spacing of transverse reinforcement (in.); $\beta$ is factor indication ability of diagonally cracked concrete to transmit tension and shear, ($\beta = 2.0$ herein); $A_v$ is area of shear reinforcement within a distance s (in$^2$); $V_p$ is component of prestressing force in the vertical direction; and $\alpha$ is angle of inclination of transverse reinforcement to longitudinal axis (degrees); $f_c'$ is the compressive strength of concrete (MPa); $b_v$ is the width of the web (mm); $f_y$ is the tensile strength of the stirrups (MPa).

### 4.2. Chinese Code 2018

The Chinese Code 2018 gives the following design formula for estimating the shear capacity of concrete beams:

$$V_d = V_{cs} + V_{sb} + V_{pb} + V_{pb,ex} \tag{5}$$

where:

$$V_{cs} = \alpha_1 \alpha_2 \alpha_3 0.45 \times 10^{-3} bh_0 \sqrt{2 + 0.6P\sqrt{f'_{cu'k}}} (\rho_{sv} f_{sv} + 0.6\rho_{pv} f_{pv}) \tag{6}$$

$$V_{pb,ex} = 0.75 \times 10^{-3} \sum \sigma_{pe,ex} A_{ex} \sin\theta_{ex} \tag{7}$$

$$V_{sb} = 0.75 \times 10^{-3} f_{sd} \sum A_{sb} \sin\theta_s \tag{8}$$

$$V_{pb} = 0.75 \times 10^{-3} f_{pd} \sum A_{pb} \sin\theta_p \tag{9}$$

where $V_d$ = shear strength (kN); $V_{cs}$ = shear strength of concrete and stirrups together (kN); $V_{sb}$ = shear strength of reinforced bar with bending (kN) ($V_{sb}$ = zero herein); $V_{pb}$ = shear strength of internal prestressing tendon with bending (kN) ($V_{pb}$ = zero herein); $V_{pb,ex}$ = shear strength of external prestressing tendon with bending (kN); $\alpha_1$ = influence coefficient of different number of bending moment ($\alpha_1$ = 1.0 herein); $\alpha_2$ = improving coefficient of prestressing ($\alpha_2$ = 1.25 herein); $\alpha_3$ = influence coefficient of compressed flange ($\alpha_3$ = 1.1 herein); $b$ = minimum web width (mm); $h_0$ = effective shear depth (mm); $P$ = percentage of longitudinal tensile reinforcement in inclined section, $P = 100\rho$; $\rho = (A_p + A_s)/bh_0$, $A_p$ is the area of the tendon (mm²), and the $A_s$ is the area of the longitudinal reinforcement (mm²), if $P > 2.5$, $P = 2.5$; $f'_{cu'k}$ = cube compressive strength of concrete (MPa); $\rho_{sv}$ = ratio of stirrup and vertical prestressed reinforcement, $\rho_{pv} = A_{pv}/bs$, $\rho_{sv} = A_{sv}/bs$; $f_{sv}$, $f_{pv}$ = specified minimum yield strength of reinforcing bars and prestressing tendon (MPa); $A_{sv}$, $A_{pv}$ = area of transverse shear reinforcement and vertical prestressed reinforcement, $A_{sb}$, $A_{pb}$, $A_{ex}$ = area of bending reinforcement, internal prestressed bending reinforcement and external prestressed bending reinforcement; $\theta_s$, $\theta_p$, $\theta_{ex}$ = angle between tangent and horizontal line of bending reinforcement, internal prestressed bending reinforcement and external prestressed bending reinforcement (degrees); $\sigma_{pe,ex}$ = effective prestress of external prestressing tendon.

The average ratios of the experimental results to the predictions of AASHTO 2017 and Chinese Code 2018 are 2.12 and 1.80, with standard deviations of 0.58 and 0.44, respectively. All calculations by design codes showed a conservative prediction of shear strength for test beams.

### 4.3. Comparison between ACI 318-14 and Test Results

In the light of the ACI 318-14 [25], strut-and-tie models (STM) shall consist of struts and ties connected at nodes to form an idealized truss. In the STM, struts are the compression members, ties are the tension members, and the nodes are the joints the supports and the load are acting on and within a joints-region. The axes of the struts and ties are chosen to approximately coincide with the axes of the compression and tension fields, respectively. The cross-sectional dimensions of a strut or tie are designated as thickness and width, and both directions are perpendicular to the axis of the strut or tie. Thickness is perpendicular to the plane, and width is in the plane of the strut-and-tie model. A tie consists of external tendons.

The strengths of ties, compression struts, and joints were determined by those provisions, the minimum value of the three parts was finally selected as the predicted shear strength.

Based on the ACI 318-14, the angle ($\theta$) between the axis of any strut and any tie entering a common node not be less than 25°. Thus, beams of the shear span-depth ratio of

2.3 ($\theta$ is 24.44°) are not suitable to analyze using the strut-and-tie model. The strength of the strut can be obtained as follows:

$$F_{ns} = f_{ce} A_{cs} \tag{10}$$

where $F_{ns}$ is the strength of the strut (MPa); $f_{ce}$ is the effective compressive strength of the concrete in the strut; $A_{cs}$ is the area of the strut (mm²). According to the ACI 318-14, the effective compressive strength of the concrete could be calculated as follows:

$$f_{ce} = 0.85 \beta_s f'_c \tag{11}$$

where $\beta_s$ is the strut coefficient, which could be obtained by ACI 318-14 ($\beta_s$ = 0.75 herein); and $f'_c$ is concrete compressive strength (MPa). The results of the strength of struts based on ACI 318-14 are listed in Table 7.

**Table 7.** Strength of the struts based on ACI 318-14.

| Specimens | $l_b$ (mm) | $\theta$ (°) | $\sin \theta_s$ | $w_s$ (mm) | $b$ (mm) | $A'_{cs}$ (mm²) | $\beta_s$ | $f'_c$ (MPa) | $F_{ns}$ (kN) | $V_{ns}$ (kN) |
|---|---|---|---|---|---|---|---|---|---|---|
| M1.3-C85-P37-None-None | 150 | 38.29 | 0.62 | 183.26 | 110.00 | 20158.79 | 0.75 | 81.09 | 1042.106 | 646.11 |
| S1.3-C85-P37-N2-b280 | 150 | 38.29 | 0.62 | 183.26 | 110.00 | 20158.79 | 0.75 | 84.30 | 1083.359 | 671.68 |
| S1.3-C85-P46-N2-b280 | 150 | 38.29 | 0.62 | 183.26 | 110.00 | 20158.79 | 0.75 | 84.30 | 1083.359 | 671.68 |
| S1.8-C85-P37-N2-b280 | 150 | 29.98 | 0.50 | 174.61 | 110.00 | 19207.43 | 0.75 | 86.64 | 1060.884 | 530.44 |
| S1.3-C115-P37-N2-b280 | 150 | 38.29 | 0.62 | 183.26 | 110.00 | 20158.79 | 0.75 | 114.85 | 1475.964 | 915.10 |
| S1.3-C55-P37-N2-b280 | 150 | 38.29 | 0.62 | 183.26 | 110.00 | 20158.79 | 0.75 | 53.97 | 693.5808 | 430.02 |
| S1.8-C85-P37-N2-b420 | 150 | 29.98 | 0.50 | 174.61 | 110.00 | 19207.43 | 0.75 | 85.46 | 1046.435 | 523.22 |

Note: where $F_{ns}$ is the strength of the struts, and the $V_{ns}$ is the strength of struts projection in the vertical direction. $w_s$ is the width of the struts.

Strength of ties can be written as:

$$F_{nt} = A_{ts} f_y + A_{tp} \left( f_{se} + \Delta f_p \right) \tag{12}$$

where $A_{ts}$ = area of non-prestressed reinforcement in a tie (mm²) ($A_{ts}$ = 226.2 mm² for monolithic beam herein); $f_y$ = specified yield strength of reinforcement (MPa) ($f_y$ = 461.61 MPa herein); $A_{tp}$ = area of prestressing steel in a tie (mm²) ($A_{tp}$ = 219.2 mm² herein); $f_{se}$ = effective stress in prestressing steel (MPa); and $\Delta f_p$ = increase in stress in prestressing steel due to factored load (MPa) ($\Delta f_p$ = 70 MPa herein).

The results of strength of ties are listed in Table 8.

**Table 8.** Strength of the ties based on ACI 318-14 and modified STM.

| Specimens | $A_{ts}$ (mm²) | $f_y$ (mm²) | $A_{tp}$ (mm²) | $f_{se} + \Delta f_p$ (MPa) | $0.9 f_p$ (MPa) | $\tan \theta$ | $V_{nt}$ (kN) | $V_{mt}$ (kN) |
|---|---|---|---|---|---|---|---|---|
| M1.3-C85-P37-None-None | 226.20 | 461.50 | 219.20 | 961.00 | 1674.00 | 0.79 | 232.07 | 372.10 |
| S1.3-C85-P37-N2-b280 | 0.00 | 461.50 | 219.20 | 892.50 | 1674.00 | 0.79 | 144.14 | 289.69 |
| S1.3-C85-P46-N2-b280 | 0.00 | 461.50 | 219s.20 | 895.25 | 1674.00 | 0.79 | 144.58 | 289.69 |
| S1.8-C85-P37-N2-b280 | 0.00 | 461.50 | 219.20 | 870.00 | 1674.00 | 0.58 | 127.16 | 211.70 |
| S1.3-C115-P37-N2-b280 | 0.00 | 461.50 | 219.20 | 878.25 | 1674.00 | 0.79 | 134.78 | 289.69 |
| S1.3-C55-P37-N2-b280 | 0.00 | 461.50 | 219.20 | 834.75 | 1674.00 | 0.79 | 116.07 | 289.69 |
| S1.8-C85-P37-N2-b420 | 0.00 | 461.50 | 219.20 | 718.75 | 1674.00 | 0.58 | 104.91 | 211.70 |

Note: $V_{nt}$ is the strength of ties based on ACI 318-14. $V_{mt}$ is the calculated strength of ties based on modified STM.

Strength of nodal zones is:

$$F_{nn} = f_{ce} A_{nz} \tag{13}$$

$$f_{ce} = 0.85 \beta_n f'_c \tag{14}$$

where $A_{nz}$ = smaller of the face of the nodal zone or a section through a nodal zone (mm²); and $\beta_n$ = factor to account for the effect of the anchorage of ties on the effective compressive strength of a nodal zone ($\beta_n$ = 0.8 herein); $f_{ce}$ = the effective compressive strength of the concrete in the nodal zones (MPa).

Tables 7–9 show the calculation results of each specimen using the strut-and-tie model based on the ACI 318-14 Appendix A provisions. For segmental specimens with high strength concrete ($f'_c$), the average of the test value of shear strength to the predicted shear strength was 1.74 with a standard deviation of 0.24. It was indicated that the strut-and-tie model on ACI 318-14 underestimated the experimental results.

**Table 9.** Summary of the calculated results.

| Specimens | Calculated Results Based on ACI 318-14 | | | Calculated Results Based on Modified STM | | | | | |
|---|---|---|---|---|---|---|---|---|---|
| | $V_{ns}$ (kN) | $V_{nt}$ (kN) | $V_{min}$ (kN) | $V_{ns}$ (kN) | $V_{ts}$ (kN) | $V_{min}$ (kN) | Control Factors | $V_{test}$ (kN) | $V_{test}/V_{min}$ |
| M1.3-C85-P37-None-None | 646.11 | 232.07 | 232.07 | 340.47 | 372.10 | 340.47 | Struts | 364.50 | 1.07 |
| S1.3-C85-P37-N2-b280 | 671.68 | 144.14 | 144.14 | 289.40 | 289.69 | 289.40 | Struts | 243.75 | 0.84 |
| S1.3-C85-P46-N2-b280 | 671.68 | 144.58 | 144.58 | 289.40 | 289.69 | 289.40 | Struts | 286.00 | 0.99 |
| S1.8-C85-P37-N2-b280 | 530.44 | 127.16 | 127.16 | 193.45 | 211.70 | 193.45 | Struts | 180.05 | 0.93 |
| S1.3-C115-P37-N2-b280 | 915.10 | 134.78 | 134.78 | 394.27 | 289.69 | 289.69 | Ties | 278.00 | 0.96 |
| S1.3-C55-P37-N2-b280 | 430.02 | 116.07 | 116.07 | 185.28 | 289.69 | 185.28 | Struts | 181.50 | 0.98 |
| S1.8-C85-P37-N2-b420 | 523.22 | 104.91 | 104.91 | 190.81 | 211.70 | 190.81 | Struts | 201.10 | 1.05 |

For all beams calculated based on ACI 318-14: the average of $V_{test}/V_{min}$ is 1.74, with the STDEVA being 0.24. For all beams calculated based on modified STM: the average of $V_{test}/V_{min}$ is 0.98, with the STDEVA being 0.08.

Note: $V_{min}$ is the minimum value of the $V_{ns}$ and $V_{nt}$.

This section may be divided by subheadings. It should provide a concise and precise description of the experimental results, their interpretation, as well as the experimental conclusions that can be drawn.

## 5. Recommendation

*Shear Capacity Equation Based on Modified STM*

According to the above calculation and analysis in Section 4.3, it can be conducted that the strength of the concrete strut is significantly greater than that of the tie. The strength of ties is the controlling factor when calculating the shear capacity of precast high-strength concrete segmental beams. However, the concrete failure of concrete in the shear-compressive zone is observed in the tests. The failure of the strut in several PCSBs is prior to that of the tie. The failure patterns are different from the above calculation conclusion.

Furthermore, the value of $\Delta f_p$ in calculation is much smaller than test value; this is the other reason that led to the smaller prediction of shear bearing capacity by those formulae. Therefore, the modified strut-and-tie model (STM) is proposed and verified in this section.

The computational model based on the ACI 318-14 Appendix A provision is shown in Figure 12. The cross-section area of the strut is listed as follows:

$$A'_{cs} = b \times l_b \sin\theta \tag{15}$$

Strength of the struts can be calculated as follow:

$$F_{ms} = 0.85\beta_s f'_c \times A'_{cs} \tag{16}$$

where $A'_{cs}$ is the modified area of the struts (mm$^2$).

Table 4 shows the value of the external tendons' stress when the failure occurs to the beams.

It can be obtained that the increment in the stress of the prestressing almost reaches the ultimate strength of the external tendons when the load reaches the shear capacity. Thus, the stress in tendons in failure ($f_{se} + \Delta f_p$) is taken as $0.9f_p$; $f_p$ is the tensile strength of strands. The strength of ties in this paper can be calculated as:

$$F_{mt} = A_{ts}f_y + 0.9A_{tp}f_p \tag{17}$$

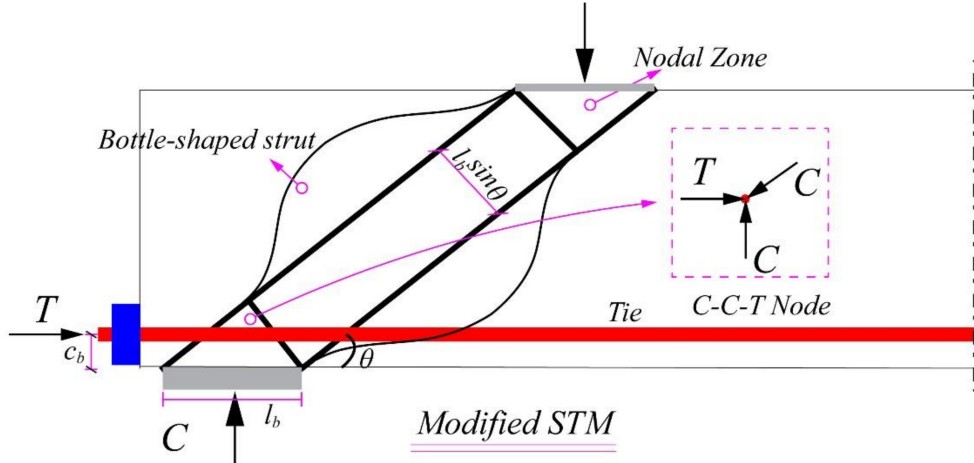

**Figure 12.** Computational model of the modified STM.

The results of the strength of ties are listed in Table 8.

In the precast concrete segmental beams, the dry joints are always located between the support and the load points. The opening of the dry joints is observed in the test when the load approaches the ultimate load. According to the Jiang [16], the shear capacity of monolithic beams is significantly higher than the precast concrete segmental beams. Considering the shear strength reduction of the precast concrete segmental beams specifically, the strength reduction factors $\varnothing_v$ is suggested by AASHTO (2003) [9] to calculate the shear capacity of the precast concrete segmental beams with dry joints. The reduced shear strength of the struts can be obtained as follows:

$$F'_{ms} = \varnothing_v 0.85 \beta_s f'_c \times A'_{cs} \tag{18}$$

where the $\varnothing_v$ is the strength reduction factors for precast concrete segmental beams according to AASHTO (2003).

Based on the AASHTO (2003), $\varnothing_v$ is 0.85 when the precast conventional concrete segmental beams with the dry joints and unbonded tendons. However, a strength reduction factor of 0.85 is set for conventional concrete. In this paper, the strength reduction factor of 0.85 is also extended to calculate the shear capacity of the precast high-strength concrete segmental beams.

The shear capacity of the specimens is the smaller values of strength of strut, ties, or nodal zone. The strength of the nodal zone is significantly greater than that of the struts and ties. Therefore, the strength of the nodal zone is not discussed in this paper.

The summary of the ACI 318 -14 and modified STM is listed in Table 9. The ratio of the test result to the calculated value based on the modified STM and the average standard deviation are 0.98 and 0.08, respectively, which indicates that the strength reduction factor of 0.85 according to AASHTO (2003) is suitable to calculate the shear capacity of precast high strength segmental beams with external tendons and dry joints.

## 6. Conclusions

Experimental studies of nine beams with T cross-sections were conducted to investigate the effect of concrete strength, shear span-depth ratio, stirrup ratio, joint number, and joint location on the shear behavior of ED-PHCSBs. Based on the test results, the following conclusions can be observed:

1. For monolithic beams, the flexural cracks were initiated first at the bottom of the web. The initial web-shear cracks of the segmental beams occurred from the root of the shear keys. These web cracks propagated along the line linking the loading point and the support, and a diagonal primary crack caused the shear failure of the monolithic

and segmental beams. The higher the concrete strength and the higher the stirrup ratio, the fewer cracks were observed in the segmental beams when failure occurred.

2.  The tension stress in external tendons kept almost constant until the opening of the joints. After joints opened, the tension stress increased rapidly while the increasing rate of the load is descended.

3.  The segmental beams with external tendons and dry joints reduced the shear strength by about 30% compared to the monolithic beam. Increasing the strength of concrete or stirrup ratio can effectively improve the ultimate shear capacity of the precast high-strength segmental beams. The shear span-depth ratio is inversely proportional to ultimate shear capacity for all specimens.

4.  The average ratios of the experimental results to the predictions by AASHTO 2017 and Chinese Code 2018 were 2.12 and 1.80, with a standard deviation of 0.58, and 0.44, respectively. AASHTO 2017 calculations show an over-conservative estimation of shear strengths for test beams. When it was extended to calculate the shear strength of ED-PHCSBs, the calculated results of Chinese Code 2018 were closer to the experimental results than those by AASHTO 2017.

5.  The ACI 318-14 Appendix A provisions predicted the shear strength of precast high-strength concrete segmental beams conservatively; the average ratio of experimental shear strength to the calculated value was 1.74 with a standard deviation of 0.24. The calculation value of the concrete struts in ACI 318-14 Appendix A provision is significantly larger than the ties. The failure controlling factor of the test beams is determined by the controlling factor of ties. It was contradicted by the fact that some test beams were destroyed by the concrete shear compression.

6.  The modified STM presented in this paper comprehensively considered two failure modes of concrete shear failure and steel strand fracture. The reduction factor of 0.85 for the effect of dry joints based on AASHTO 2003 was adopted to predict the shear capacity of ED-PHCSBs. The average ratios of experimental shear strengths to the calculated values by modified STM were 0.98 with a standard deviation of 0.08. It is indicated that the strength reducing factor of 0.85 suiting for the conventional concrete also calculate accurately to the ultimate shear strength of ED-PHCSBs.

**Author Contributions:** Methodology, Z.H.; formal analysis, Z.H.; investigation, Y.C.; writing—original draft preparation, Z.H. and Y.C.; writing—review and editing, H.J., J.X., Z.X. and S.Z.; supervision, H.J.; project administration, H.J. All authors have read and agreed to the published version of the manuscript.

**Funding:** This research received no external funding or This research was funded by [National Natural Science Foundation of China] grant number [No.51778150, NO.51808133], [Natural Science Foundation of Guangdong Province in China] grant number [2016A030313699], [the Science and Technology Planning Project of Guangzhou City in China] grant number [No.201804010422] And The APC was funded by [National Natural Science Foundation of China].

**Conflicts of Interest:** The authors declare no conflict of interest.

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
