# Peer review of "Experimental Study on Shear Behavior of Precast High-Strength Concrete Segmental Beams with External Tendons and Dry Joints"

_buildings, doi:10.3390/buildings12020134_

Round 1
Reviewer 1 Report
| Comments to Author: |
|---|
| There are shortcomings of the manuscript which fall short of the required standard for publication in a top-rated journal such as Build. |
Reviewer 2 Report
Precast high-strength concrete segmental beams with external tendons and dry joints are a useful alternative for achieving the accelerated bridge construction, due to the lighter self-weight and the easier installation. The authors have carried out detailed experimental work and collected lots of data. Further, the authors compare the test results with a strut and tie model. However, they do not show the derivation of the model. Please show this in a simple manner for the understanding of the readers.
Over, this is a good work.
Reviewer 3 Report
Reviewer's opinion
The article deals with the experimental study on shear behavior precast high-strength concrete segmental beams with external tendons and dry joints. The article has a wide and interesting experimental campaign in which different parameters that influence shear resistance are studied. In addition, the experimental results are compared with the predictions of the AASHTO 2017 and Chinese Code 2018 regulations. The strut-and-tie model of the ACI 318-14 recommendation is also used, which according to the experimental results, the authors propose a modification of the same more realistic.
The article is relevant to the readers of Buildings. It is well written, with adequate analysis for scientific research. Thus, the reviewer proposes its acceptance for publication in Buildings
Nevertheless, the reviewer has the following minor suggestions:
Specify the number of characterization specimens of the tested concretes and put the mean values and standard deviation (Table 2)
Explain in more detail the process of pouring concrete into the beams.
Line 205: use of capital letter (The)
In Fig. 5, what does Vu mean?
Lettering subfigure graphs (like Figure 6 and 8)
